# Comparative Analysis and Validation of the IMPEDED VTE, IMPEDE VTE, and SAVED Risk Models in Predicting Venous Thromboembolism in Multiple Myeloma Patients: A Retrospective Study in Türkiye

**DOI:** 10.3390/diagnostics15050633

**Published:** 2025-03-05

**Authors:** Vildan Gursoy, Mehmet Baysal, Sevil Sadri, Fazil Cagri Hunutlu, Tuba Ersal, Ozgur Omer Gul, Elif Kose, Esra Celik, Serap Baysal, Tuğba Gullu Koca, Sinem Cubukcu, Ezel Ergun, Seyma Yavuz, Vildan Ozkocaman, Fahir Ozkalemkas

**Affiliations:** 1The Division of Hematology, The Department of Internal Medicine, The Faculty of Medicine, Uludag University, 16059 Bursa, Türkiye; 2The Department of Hematology, Tekirdag İ. Fehmi Cumalioglu City Hospital, 59030 Tekirdag, Türkiye; drmehmetbaysal@gmail.com; 3The Division of Hematology, The Department of Internal Medicine, Bursa City Hospital, 16250 Bursa, Türkiye; 4The Department of Internal Medicine, Bursa City Hospital, 16250 Bursa, Türkiye; ozguromergul@gmail.com (O.O.G.);; 5The Department of Internal Medicine, The Faculty of Medicine, Uludag University, 16059 Bursa, Türkiye; 6The Department of Public Health, The Faculty of Medicine, Tekirdag Namık Kemal University, 59030 Tekirdag, Türkiye

**Keywords:** IMPEDE VTE, IMPEDED VTE, multiple myeloma, thrombotic risk scores, venous thromboembolism

## Abstract

**Background:** Several thrombotic risk assessment models have been proposed for identifying patients with a high risk of thrombosis (the IMPEDE venous thromboembolism (VTE), SAVED, and PRISM scores) in multiple myeloma (MM). Recently, adding a biomarker (D-dimer) for the IMPEDE VTE score has shown that it can boost the detection power of IMPEDED VTE. However, data from studies comparing these models in MM are scarce. Even real-world data arguing the utility of thrombotic risk assessment models in MM from low- or middle-income countries like Türkiye are lacking. **Methods:** We aimed to show the possibility of detecting VTE using the IMPEDED VTE score in our cohort by retrospectively screening MM patients. Therefore, we aimed to compare the IMPEDE VTE, SAVED and IMPEDED VTE scoring models. **Results:** We conducted a retrospective analysis of 455 MM patients from three centers in Bursa, Türkiye, evaluating the incidence of VTE within six months of the treatment. The IMPEDED VTE score showed superior predictive accuracy (c-statistic of 0.701), compared to the IMPEDE VTE (0.618) and SAVED (0.633) scores, demonstrating the added value of D-dimer as a biomarker. The cumulative incidence of VTE in the cohort was 10.7%, comparable to rates observed in real-world studies. **Conclusions:** Despite the interventions and thrombotic risk assessment models, thrombosis remains a high-risk entity. Personalized risk assessment tools, such as IMPEDED VTE, could be used to manage thrombotic risk in MM patients, particularly in resource-limited settings. Albeit the thromboprophylaxis (51.6%), our findings support the utility of biomarker-enhanced models for better VTE-risk stratification, particularly in resource-limited settings.

## 1. Introduction

Multiple myeloma (MM) is a hematological malignancy accounting for approximately 10–15% of all blood cancers [1]. A significant complication in MM patients is the increased risk of venous thromboembolism (VTE), including both deep vein thrombosis (DVT) and pulmonary embolism (PE) [2]. The identification and management of VTE in MM patients is of crucial importance due to the substantial morbidity and mortality associated with thrombotic events [3]. An increased risk of VTE has been shown in previous studies after the invention of immunomodulatory drugs (IMIDs), both in clinical trials and real-world data [4,5,6]. In a population-based cohort study, it has been shown that VTE is associated with a 67% increased relative risk of mortality in six months in newly diagnosed MM patients [7]. The thrombotic risk in MM is not only caused by disease-related factors, comorbidities, and patient-related factors, but also comes from the medications used to treat MM, such as IMIDs (thalidomide, lenalidomide, pomalidomide, etc.), high dose dexamethasone, and doxorubicin [2,4,6,8].

In a study performed to define thrombotic risk and thromboprophylaxis in 2008, Palumbo et al. proposed the first thrombotic risk assessment in MM [9]. The study was a comprehensive risk assessment involving disease-related, patient-related, and treatment-related risk factors. This risk classification has also put various recommendations into effect. While the use of acetylsalicylic acid was recommended with a score of 1≤, prophylactic low-molecular-weight heparin (LMWH) or therapeutic vitamin K antagonists were proposed with a score of 2≥ for high-risk groups [9].

Additionally, several risk models have been proposed to identify MM patients at high risk due to thrombosis. The IMPEDE VTE, SAVED, and PRISM scores are among the most recognized models [10,11,12]. These models incorporate various clinical and laboratory parameters to stratify patients based on their thrombotic risks. The IMPEDE VTE score is composed of the following nine variables: IMIDs; body mass index; pelvic, hip, or femur fracture; use of erythropoiesis-stimulating agents; use of dexamethasone/doxorubicin; ethnicity/race; history of VTE; tunneled-line central venous catheters; and existing thromboprophylaxis. Even so, the SAVED score comprises the following five variables: surgery, Asian ethnicity, history of VTE, eighty years of age, and dexamethasone. The PRISM score also involves prior VTE, ethnicity, IMID, surgery, and metaphase cytogenetics [10,11,12]. The most important difference between the PRISM score and others is the use of a genetic marker (metaphase cytogenetics) in the risk classification of thrombosis [12]. Unfortunately, the interphase fluorescence in situ hybridization (iFISH) score is the accepted method worldwide in the genetic risk classification of plasma cells in MM. Cytogenetic evaluation of plasma cells, especially metaphase cytogenetics, is, unfortunately, unlikely to be performed in all MM patients. Recently, an enhanced version of the IMPEDE VTE score, also known as the IMPEDED VTE score, has been developed by incorporating a specific biomarker for thrombosis (D-dimer) [13]. The addition of D-dimer has contributed to improving the detection power for VTE in MM patients, potentially allowing for better prophylactic and therapeutic strategies [13].

Therefore, the present study aimed to evaluate the effectiveness of the IMPEDE VTE, IMPEDED VTE, and SAVED scores in our cohort of MM patients by conducting retrospective screening. We sought to determine the added value of the biomarker-enhanced score in predicting the risk of VTE in Turkish MM patients. This comparison will help assess whether the IMPEDED VTE score provides a superior predictive capability and could be integrated into routine clinical practice to manage better the thrombotic risks in MM patients. Furthermore, as there is no independent validation of the IMPEDED VTE score, with this analysis, we aimed to provide an external validation of the IMPEDED VTE score in Turkish patients.

## 2. Methods

In this retrospective cohort study, we evaluated patients diagnosed and treated due to MM in three facilities between 2019 and 2023 in Bursa, Türkiye. Patients with another type of cancer, those with missing data, or those not receiving treatment for active MM were excluded from our analysis. The data were obtained from the electronic medical records and files of the patients in the hospital. A total of 567 patients were screened in our study, and the patients with no D-dimer value before receiving active treatment for MM, those starting the therapy for VTE before receiving treatment for MM, those admitted to other facilities for the treatment after the diagnosis, or those lost during the follow-ups were also excluded from the criteria. Therefore, a total of 455 patients were included in our analysis. A detailed flow diagram of the patient’s selection process is highlighted in Figure 1. VTE was defined as having symptomatic or incidental, new or recurrent, PE, proximal or distal lower extremity DVT, or proximal or distal upper extremity DVT.

Such characteristics as demographic data, treatment details, age, gender, calculation of BMI, recent surgical history of comorbidities, recent fractures, use of medications, and anti-platelet and anti-thrombotic medications were screened and analyzed from the medical files. We calculated the IMPEDE VTE score in light of the following points: use of IMIDs (+4 points), BMI ≥ 25 kg/m^2^ (+1 point), recent pelvic, hip, or femur fractures (+4 points), erythropoiesis-stimulating agent (+1 point), doxorubicin (+3 points), dexamethasone (+4 and +2 points for high- and low-doses, respectively, high dose > 160 mg/month and low dose <160 mg/month), Asian/Pacific islander race (−3 points), prior history of VTE (+5 points), CVC (+2 points), use of therapeutic low-molecular-weight heparin or warfarin (−4 points), and use of prophylactic low-molecular weight acetylsalicylic acid (−3 points).

Samples of venous blood were drawn and placed in sodium citrate-containing tubes to test D-dimer. D-dimer levels (µg/mL fibrinogen equivalent units [FEU]) were measured by a sandwich enzyme-linked immunosorbent assay (ELISA) (Diagnostica Stago). The cut-off value for the upper limit of the d-dimer level was 0.5 µg/mL. We have also stratified D-dimer levels based on the study, suggesting the IMPEDE VTE risk score as follows: −2 points for D-dimer < 0.41 µg/mL, −1 point for D-dimer between 0.41 and <0.83 µg/mL, 0 points for D-dimer between 0.83 and <1.70 µg/mL, 1 point for D-dimer between 1.70 and <3.31 µg/mL, and 2 points for D-dimer ≥3.31 µg/mL. The following five parameters were used to calculate the SAVED model: 80 years of age or older (+1 point), Asian etchnicity (−3 points), prior history of VTE (+3 points), major thoracic, neurological, orthopedic, abdominal, or urological procedures within the previous 90 days (+2 points), and dexamethasone dose over a one-month course (+2 points for high dose, defined as >160 mg, +1 point for standard dose, defined as 120–160 mg, or 0 points for low dose, defined as <120 mg). All points were calculated to acquire a total score. For the SAVED score, while between 0 and 1, the points were classified as having low risk, ≥2 points were classified as having high risk. The IMPEDE VTE or IMPEDED VTE scores of ≤3, 4–7, and ≥8 were considered to have low risk, intermediate risk, and high risk, respectively (detailed variables and interpretation of the scoring models are demonstrated in Table 1).

High-risk cytogenetics related to iFISH were regarded as (4;14), t(14;16), t(14;20), del(17/17p), and gain(1q) [14]. A recent fracture was defined as pelvic, femoral, or hip fractures occurring between the last 30 days of MM diagnosis and the start of systemic therapy. Prophylactic and therapeutic enoxaparin doses were set at 40 mg per day and 1 mg/kg twice per day. The treatment for MM consists of an induction regimen and autotransplantation in transplant-eligible patients. The induction regimen consists of a doublet, triplet, or quadruplet, depending on the patient’s characteristics, transplant eligibility, and fitness level.

The baseline characteristics of individuals with and without VTE within 6 months after the initiation of medication were descriptively evaluated. The continuous variables were reported as median with interquartile range (IQR), and the medians between the two groups were compared using the Wilcoxon-Mann–Whitney test. However, the categorical variables were compared using the chi-square or Fisher’s exact tests. The cumulative incidence of VTE was estimated using the Kaplan–Meier method with death as a competing risk. The analysis of the Receiver Operating Characteristic curve (ROC) was performed and reported as a c-statistic to assess the discriminative power of the IMPEDE VTE, IMPEDED VTE and SAVED scores to predict VTE within the last 6 months of the initiation of the treatment. Harrell’s C-index was used to measure external model discrimination, and 500 resampling bootstraps were used to calculate a 95% CI. The model’s calibration was evaluated using the Kaplan–Meier risk categories with the current dataset. All methods were performed under the relevant guidelines.

## 3. Results

Our final cohort included 455 patients. Of the 455 patients, 266 (58.5%) were female and 189 (41.5%) were male. The median age of the study population was 64 years, ranging from 30 to 90. The immunoglobulin G (IgG) was the most frequently detected subtype in 222 (49.8%) patients. IgA, light chain, multiple plasmacytoma, IgM, IgD, and asecretory types were, respectively, detected as 94 (20.7%), 113 (24.8%), 11 (2.4%), eight (1.8%), five (1.1%) and two (0.4%). The International Staging System (ISS) scores, however, were detected to be I, II, III, and unknown 94 (20.7%), 142 (31.2%), 199 (43.7%), and 20 (4.4%), respectively. The Revised ISS (R-ISS) scores were also I, II, III, and unknown 79 (17.4%), 257 (56.5%), 94 (20.7%), and 25 (5.5%), respectively. Fifty-five (12.1%) patients had high-risk FISH cytogenetics, 312 (68.6%) had normal iFISH results, and 63 (13.9%) had del(13q), while 25 (5.5%) patients had unknown or missing iFISH results. The baseline, demographic, clinical, and treatment characteristics of the patients are presented in Table 2.

There were no patients with Asian or Pacific islander ethnicity. BMI was found to be >25 kg/m^2^ in 282 (61.9%) patients. Forty-six (10.1%) patients had pelvic or hip fractures. While 7 patients (1.5%) had major surgery within the previous 90 days of MM diagnosis, 148 patients (32.6%) had a central venous catheter. Seven patients (1.5%) also had a history of VTE. Seventeen patients (3.7%) were 80 years of age. Six patients (1.3%) were on erythropoiesis-stimulating agents. The monthly dexamethasone dose was found to be >160 mg in 316 patients (69.4%), while the dexamethasone dose was detected as <160 mg in 139 patients (30.6%). Twenty-two (4.8%) patients were detected to receive doxorubicin. While 220 patients (48.3%) used no anti-aggregators or anti-coagulants, 215 patients (47.3%) were on low-dose acetylsalicylic acid. Twenty patients (4.4%) were on therapeutic low-molecular-weight heparin or vitamin K antagonists.

The majority of the patients were treated with bortezomib-based triplet therapy in the first-line setting. In total, 295 (64.8%), 57 (12.5%), 22 (4.8%), and 1 (0.2%) patients were treated with VCD (bortezomib, cyclophosphamide, dexamethasone), VRD (bortezomib, lenalidomide, dexamethasone), VAD (bortezomib, doxorubicin and dexamethasone), and VTD (bortezomib, thalidomide, dexamethasone), respectively. Even so, while 74 patients (16.3%) were treated with bortezomib dexamethasone, 2 (0.4%) patients were treated with lenalidomide dexamethasone as first-line doublet therapy. In addition, four patients (0.9%) completed first-line treatment with quadruplet therapy with DRVd (bortezomib, daratumumab, lenalidomide, and dexamethasone). Generally, several combined agents are used to provide induction in the initial setting of MM treatment, while also considering frailty. We have grouped our patients according to the induction regimen used. In total, 375 patients received a triplet regimen in the first line; 295 (64.8%), 57 (12.5%), 22 (4.8%), and 1 (0.2%) patients were treated with VCD (bortezomib, cyclophosphamide, dexamethasone), VRD (bortezomib, lenalidomide, dexamethasone), VAD (bortezomib, doxorubicin, and dexamethasone), and VTD (bortezomib, thalidomide, dexamethasone), respectively. Eighty patients were treated with a doublet regimen. Seventy-four patients (16.3%) were treated with bortezomib dexamethasone, and two (0.4%) patients were treated with lenalidomide dexamethasone as first-line doublet therapy. In addition, four patients (0.9%) completed first-line treatment with quadruplet therapy, including an anti-CD38 monoclonal antibody with DRVd (bortezomib, daratumumab, lenalidomide, and dexamethasone).

Under the SAVED thrombosis risk scoring system, 184 (40.4%) patients were classified in the low-risk group and 271 (59.6%) in the high-risk group. Given the IMPEDE VTE thrombosis risk classification, 158 (34.7%), 221 (48.6%), and 76 (16.7%) patients were evaluated as low, intermediate, and high-risk groups, respectively. According to the IMPEDED VTE risk classification, however, 152 (33.4%), 212 (46.6%), and 91 (20%) patients were evaluated to be at low, intermediate, and high risk, respectively. The median D-dimer value of the study participants was found to be 1.21 µg/mL fibrinogen equivalent units (FEU), and the upper limit of the assay was 0.5s µg/mL FEU.

A total of 32 VTE events occurred in the study period. Of the 32 VTE events, 21 (65.6%), 8 (25%), and 3 (9.4%) were lower extremity deep vein thrombosis, PE, and upper extremity thrombosis. The median time from the diagnosis of MM to a VTE event was 97 days (range: 7–178 days). The six-month cumulative incidence of VTE within the whole study period was detected as 10.7%.

The 6-month cumulative incidence of VTE regarding the IMPEDE VTE score was 3.0 [95% confidence interval (CI): 2.1–7.9] in the low-risk group, compared to 9.1% (95% CI: 8.9–16.4) and 10% (95% CI: 12.2–26.1) in the intermediate and high-risk groups. The 6-month cumulative incidence of VTE in terms of the SAVED score was 3.4% (95% CI: 2.2–5.5) for the low-risk and 10.8% (7.6–14.3) for the high-risk groups. Given the IMPEDED VTE score, the 6-month cumulative incidence of VTE was 3.4% (95% CI: 2.3–4.7), 8.5% (95% CI: 6.7–10.8), and 17.2% (95% CI: 14.3–24.1) for the low-risk, intermediate risk, and high-risk groups, respectively. The cumulative incidence of thrombosis of MM thrombotic risk models is presented in Figure 2, Figure 3 and Figure 4.

The index of c-statistics was detected as 0.701% (95% CI: 0.616–0.784), 0.633% (95% CI: 0.544–0.723), and 0.618% (95% CI: 0.529–0.706) for the IMPEDED VTE, SAVED and IMPEDE VTE scores, respectively. The detailed comparison of the MM-specific thrombosis models is also demonstrated in Table 3.

We have also analyzed the performance of the IMPEDED VTE score categorically for external validation. Each increase in the IMPEDED VTE score category was associated with a significantly higher risk of VTE: the 6-month cumulative incidence rates were 3.4% in the low-risk group (reference), 8.5% in the intermediate-risk group (HR: 3.57, 95% CI 1.13–9.77), and 17.2% in the high-risk group (HR: 12.89, 95% CI: 11.33–19.75), as detailed in Table 4.

## 4. Discussion

Despite the thromboprophylaxis with low-dose aspirin (51.6%), which would suggest that the aspirin is not adequate to be considered as thromboprophylaxis in this patient population, we have found the cumulative incidence of VTE as 10.7% within 6 months in this real-world analysis. Previous real-world studies and clinical trials demonstrated an increased risk of thrombosis, especially within the first year of diagnosis of MM [16]. Our findings are comparable and similar to those reported in previous studies and the Myeloma XI trial [17]. Despite the advances in the risk assessment of VTE in MM and thrombophylaxis regarding the risk assessment models, the risk and incidence of thrombosis continue in a certain group of patients [18,19,20]. Even adding a biomarker of VTE could boost the ability of the risk model to predict VTE. Theoretically, an ideal risk model would have a c-index of 1.0 and be able to anticipate the outcomes. A c-index of 0.5 is no better than randomizing. Both the IMPEDE VTE and SAVED models have c-indices of 0.68 and 0.74 in a real-world cohort, which is still a good indicator of prognosis [15,16]. In our analysis, the c-index for the IMPEDED VTE model was 0.70, which is comparable with real-world data. In our study, the thrombotic risk assessment models were demonstrated to be comprehensive in evaluating thrombotic risks and mitigating the impact of VTE in MM.

With the emergence of new and more effective drugs, the treatment of MM has been exposed to changes from the past to the future. While triple therapies were recommended for newly diagnosed MM patients in the first-line setting, after the effectiveness of anti-CD 38 monoclonal antibodies, quadruplet therapies are now recommended in the guidelines for the first-line MM patients in both transplant-eligible and ineligible settings [21]. However, in lower and middle-income countries with limited supply, such as Türkiye, these developments fall a little behind those in the developed countries. Furthermore, 375 patients (82.4%), the majority of whom (295 patients, 64.8%) were on VCD regimens, started receiving their first-line treatment with triple therapy. Again, the number of patients using any immunomodulatory drug in first-line treatment was 64 (14.0%), which could partly explain the low percentage (51.6%) of VTE prophylaxis in our cohort compared to real-world data [22]. In addition, in a post hoc analysis of the GRIFFIN study, VTEs were reported to develop in 10.1% of the patients treated with daratumumab plus lenalidomide/bortezomib/dexamethasone (D-RVd) and 15.7% of those receiving RVd despite thromboprophylaxis, according to the guideline proposed by the International Myeloma Working Group (IMWG) [23]. We can argue that adding a monoclonal antibody (daratumumab) does not elevate the thrombotic risk of MM; however, it demonstrates the need for a better assessment and management in stratifying the risk of thrombosis and distinguishing patients at high risk for thrombosis [23].

In a recent study, where 250 patients with MM were compared in terms of the effectiveness and safety of the three thrombosis risk models (IMPEDE VTE, SAVED, and PRISM), the majority of the patients were on low-dose ASA for thromboprophylaxis, and VTE within the 6 months was reported as 8% [22]. In this study, the authors indicated that the IMPEDE VTE score, unlike the SAVED and PRISM scores, distinguished the intermediate and high-risk groups better [22]. In another, similar, study performed in Brazil with 131 MM patients [24], the following criteria were reported: area under the curve (AUC) was 0.80% (95% CI: 0.66–0.95, *p* = 0.002) for the IMPEDE VTE score, 0.69% (95% CI; 0.49–0.89, *p* = 0.057) for the SAVED score, and 0.68% (95% CI: 0.48–0.88, *p* = 0.075) for the IMWG risk score [24]. The researchers in the study concluded that among Brazilian patients receiving IMID therapy, IMPEDE VTE was the most reliable predictor for VTE [24].

In the original study that led to the development of the IMPEDE VTE score, there was a clearer distinction in thrombotic risk between the intermediate and high-risk groups [11]. However, our study did not reflect this difference. This discrepancy might be due to factors such as differences in patient demographics, treatment regimens, the lower use of IMIDs, and the smaller sample size, which could inherently reduce the observed number of VTE events.

The results of our study support the utility of the IMPEDED VTE score, particularly the benefit of including D-dimer as a biomarker to improve the risk prediction for VTE in MM patients. This aligns with biomarker-enhanced models, such as the IMPEDED VTE score, to provide a more precise risk stratification [13]. The comparable results of the IMPEDE VTE and SAVED models with previous studies highlight that despite their limitations, these models are also valuable in clinical practice [16,22].

Furthermore, our manuscript has the potential to externally validate the IMPEDE VTE score, external validation of the IMPEDE VTE, and SAVED models, demonstrating their effectiveness in assessing VTE risk in patients with MM. The IMPEDE VTE model demonstrated robust performance, providing reliable risk differentiation across different populations [15]. Meanwhile, the SAVED score designed specifically for patients on immunomodulatory drugs (IMIDs) showed strong predictive power, although it included limited information [22]. The IMPEDED VTE model, which included D-dimer as a biomarker, achieved higher predictive values (c-statistic: 0.701) compared to the other models, indicating a potential improvement in risk and usefulness settings. These results highlight the importance of standardized risk models for optimizing thromboprophylaxis strategies in patients with MM (In Table 5, there is a detailed description and comparison of validation studies of the thrombotic scoring systems in MM).

Our study has several limitations. First, it is a retrospective study relying on electronic medical records, which may have led to missing or incomplete data. For instance, we excluded patients with missing data, those lost to follow-up, and those who received treatment outside our centers, which could introduce selection bias. Additionally, while the IMPEDED VTE score includes D-dimer as a biomarker, the timing of D-dimer measurement may have varied among patients, potentially impacting its predictive accuracy. Furthermore, our study cohort consisted entirely of Turkish MM patients, limiting the ethnic diversity in the population (e.g., no patients of Asian or Pacific Islander descent were included). This could impact the generalizability of all three risk scores, which assigns significant weight to race-related variables. Finally, the study did not account for all potential confounders, such as socioeconomic status and lifestyle factors, which may have influenced the risk of VTE.

Additionally, the study was conducted in Türkiye, a middle-income country, where access to novel therapies, such as quadruplet regimens with anti-CD38 monoclonal antibodies, may differ from that in high-income countries. As a result, the treatment patterns observed here, particularly the low use of IMIDs and VTE prophylaxis, may not reflect those in settings with more advanced healthcare systems.

## 5. Conclusions

In conclusion, in our study, the IMPEDED VTE score, which incorporates D-dimer as a biomarker, was found to outperform other risk models in predicting VTE in MM patients. We would like to underline the importance of personalized risk assessment models in managing complex thrombotic risks associated with MM. Although thrombosis remains a significant complication in MM patients, especially early in the treatment, more refined risk models can help guide better prevention strategies. As treatment options for MM continue to evolve, it is also crucial to improve our understanding of how patients can be best protected from such life-threatening complications as VTE, particularly in resource-limited settings like ours.

## Figures and Tables

**Figure 1 diagnostics-15-00633-f001:**
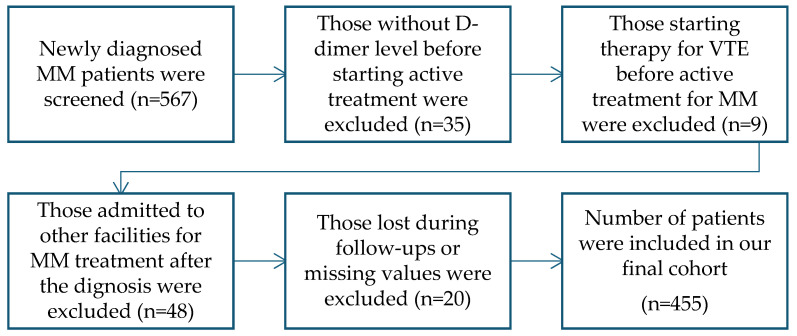
Diagram showing patient selection process. MM: Multiple myeloma, VTE: Venous thromboembolism.

**Figure 2 diagnostics-15-00633-f002:**
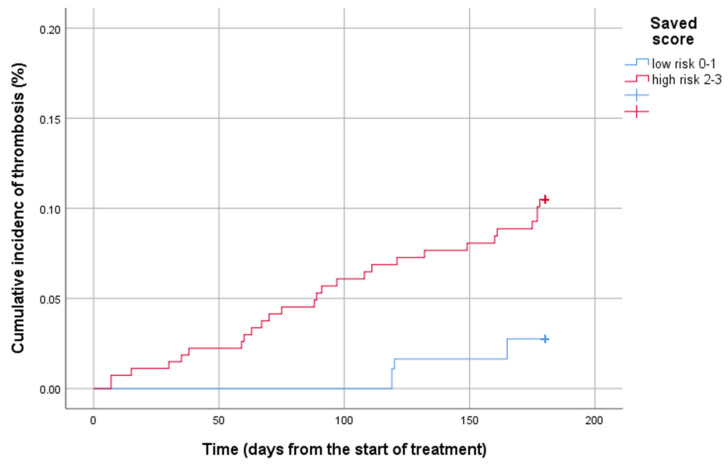
Cumulative incidence of the 6th month thrombosis by the SAVED score.

**Figure 3 diagnostics-15-00633-f003:**
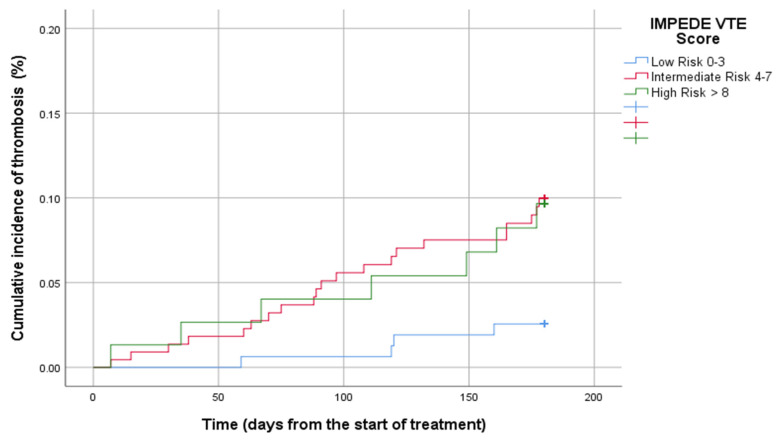
Cumulative incidence of 6th month thrombosis by the IMPEDE VTE score.

**Figure 4 diagnostics-15-00633-f004:**
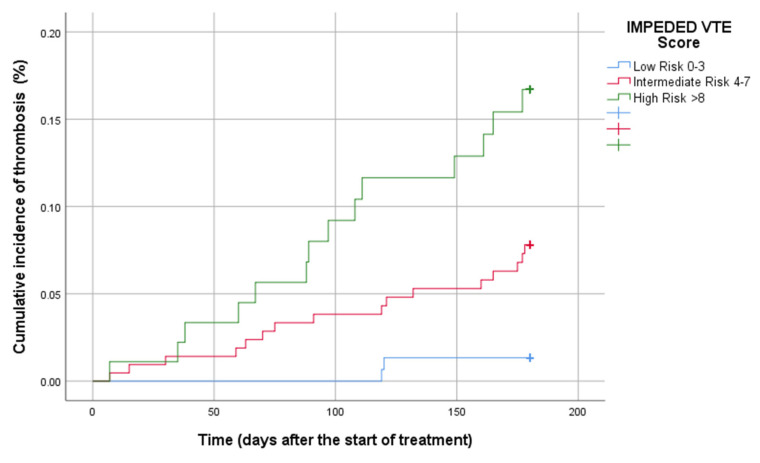
Cumulative incidence of the 6th month thrombosis by the IMPEDED VTE score.

**Table 1 diagnostics-15-00633-t001:** Detailed description and interpretation of thrombotic scoring models in multiple myeloma.

Risk Models	Factors	Scoring	Risk Classification
SAVED	Surgery	+2	Low-Risk, 0–1 High-Risk, ≥2
	Asian Ethnicity	−3	
	History of VTE	+3	
	Age ≥ 80	+1	
	Dexamethasone Dose (monthly)	+2 (High Dose > 160 mg), +1 (Standard Dose 120–160 mg), +0 (Low Dose < 120 mg)	
IMPEDE VTE	Immunomodulatory Drugs (IMIDs)	+4	Low-Risk, ≤3 Intermediate-Risk, 4–7 High-Risk, ≥8
	Body Mass Index (BMI ≥ 25 kg/m^2^)	+1	
	Pelvic, Hip, or Femur Fracture	+4	
	Use of Erythropoiesis-Stimulating Agents (ESA)	+1	
	Dexamethasone Dose (monthly)	+4 (High Dose > 160 mg), +2 (Low Dose < 160 mg)	
	Use of Doxorubicin	+3	
	Asian/Pacific Islander Race	−3	
	History of VTE	+5	
	Central Venous Catheter (CVC)	+2	
	Prophylactic Aspirin or Low-molecular Weight Heparin	−3 (Aspirin), −4 (Therapeutic Heparin or Warfarin)	
IMPEDED VTE	IMPEDE VTE Factors (All of the Above)	All Scores Above Application	Low-Risk, ≤3 Intermediate-Risk, 4–7 High-Risk, ≥8
	D-dimer Level	−2 (D-dimer < 0.41 µg/mL), −1 (0.41 -< 0.83 µg/mL), 0 (0.83 -< 1.70 µg/mL), +1 (1.70 -< 3.31 µg/mL), +2 (≥3.31 µg/mL)	

**Table 2 diagnostics-15-00633-t002:** Baseline, demographic, clinical, and treatment characteristics of the patients.

	Total Patients *n* = 455 (%)	6-Month VTE *n* = 32 (%)	No VTE *n* = 423 (%)	*p*-Value
Gender				
Female	266 (58.5)	16 (50)	250 (59.1)	
Male	189 (41.5)	16 (50)	173 (40.9)	0.354
Age (years) at Diagnosis (median, min–max)	63 (30–90)	62 (45–80)	64 (30–90)	0.940
Age > 80	17 (3.7)	1 (3.1)	16 (3.8)	0.764
BMI > 25 kg/m^2^	282 (61.9)	21 (65.6)	261 (61.7)	0.850
Recent Pelvic, Hip, or Femur Fractures	46 (10.1)	4 (12.5)	42 (9.9)	0.555
Major Surgery within 90 days of MM Diagnosis	7 (1.5)	2 (6.3)	5 (1.2)	0.081
History of VTE	7 (1.5)	1 (3.1)	6 (1.4)	0.405
Tunneled/Central Venous Line	148 (32.6)	10 (31.2)	138 (32.6)	0.149
Erythropoiesis-stimulating Agent	6 (1.4)	2 (6.2)	4 (1)	0.096
Monthly Dexamethasone Doses				
High-dose	316 (69.4)	28 (87.5)	288 (68.1)	
Low-dose	139 (30.6%)	4 (12.5)	135 (31.9)	0.027
Use of Doxorubicin	22 (4.8)	2 (6.3)	20 (4.7)	0.175
Existing Antiplatelet/ Anticoagulation Usage				
Low-dose ASA	215 (47.3)	12 (37.5)	203 (48)	0.495
Anticoagulation	20 (4.4)	2 (6.3)	18 (4.3)	
Subtypes of MM				
IgG	222 (48.8)	15 (46.9)	207 (48.9)	
IgA	94 (20.7)	9 (28.1)	85 (20.1)	
Light chain	113 (24.8)	5 (15.6)	108 (25.5)	
Multiple Plasmacytoma	11 (2.4)	3 (9.4)	8 (1.9)	0.197
IgM	8 (1.8)	0	8 (1.9)	
IgD	5 (1.1)	0	5 (1.2)	
Asecretory	2 (0.4)	0	2 (0.5)	
ISS				
I	94 (20.7)	5 (15.6)	89 (21)	
II	142 (31.2)	9 (28.1)	133 (31.4)	0.654
III	199 (43.7)	16 (50)	183 (43.3)	
Unknown	20 (4.4)	2 (6.3)	18 (4.3)	
R-ISS				
I	79 (17.4)	4 (12.5)	75 (17.7)	
II	257 (56.5)	19 (59.4)	238 (56.3)	0.737
III	94 (20.7)	7 (21.9)	87 (20.6)	
Unknown	25 (5.5)	2 (6.3)	23 (5.4)	
Laboratory values at Diagnosis				
Hgb g/dL (mean)	10.3 g/dL	9.6 g/dL	10.3 g/dL	0.466
White Blood Cell Count × 10^9^	7.34 × 10^9^/L	6.85 × 10^9^/L	7.38 × 10^9^/L	0.630
Thrombocyte Count × 10^9^	230 × 10^9^/L	212 × 10^9^/L	231 × 10^9^/L	0.573
First-line Induction Regimen Triplet Regimens				
VCD	295 (64.8)	15 (46.9)	280 (66.2)	
VRD	57 (12.8)	7 (21.9)	50 (11.8)	
VAD	22 (4.8)	2 (6.2)	20 (4.7)	
VTD	1 (0.2)	0	1 (2.3)	0.175
Doublet regimens				
VD	74 (16.3%)	8 (25%)	66 (15.6%)	
RD	2 (0.4%)	0	2 (4.7%)	
Quadruplet Regimens				
DVRd	4 (0.9%)	0	4 (9.4%)	
ASCT after Induction				
Yes	146 (32.1%)	8 (25%)	138 (32.6%)	
No	309 (67.9%)	24 (75%)	285 (67.4%)	0.436

**Table 3 diagnostics-15-00633-t003:** Distribution of the patients in detail by MM-specific thrombose risk models.

Thrombose Risk Models	*n* (%)	Six-Month Cumulative Incidence Rate of Thrombosis (%)
SAVED Model		
Low-risk	184 (40.4)	3.4
High-risk	271 (59.6)	10.8
IMPEDE VTE Model		
Low-risk	158 (34.7)	3
Intermediate risk	221 (48.6)	9.1
High-risk	76 (16.7)	10
IMPEDED VTE Model		
Low-risk	152 (33.4)	3.4
Intermediate risk	212 (46.6)	8.5
High-risk	91 (20)	17.2

**Table 4 diagnostics-15-00633-t004:** Summary of external validation results for IMPEDE VTE, IMPEDED VTE, and SAVED risk models in multiple myeloma patients.

Models	Validation Study	Validation Results (AUC or C-Statistics	Key Interpretation
IMPEDE VTE	Covut et al. [15]	AUC: 0.80 (95% CI: 0.66–0.95)	Reliable in distinguishing risk categories.
SAVED	Dima et al., 2023 [16]	C-statistics: 0.68	Demonstrated high predictive value.
IMPEDED VTE	Current study	C-statistics: 0.701	Biomarker-enhanced model; performs comparably to others.

**Table 5 diagnostics-15-00633-t005:** Performance of IMPEDED VTE score for external validation.

IMPEDED VTE Model *n* (%)	6th Month Cumulative Incidence Rate of Thrombosis (%)	HR for VTE (95% CI)
Low-risk, 152 (33.4)	3.4	1 (reference)
Intermediate-risk, 212 (46)	8.5	3.57 (1.13–9.77)
High-risk 91, (20)	17.2	12.89 (11.33–19.75)

## Data Availability

The datasets used and/or analyzed during the current study are available from the corresponding author upon reasonable request.

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
