# Peer review of "Comparative Analysis and Validation of the IMPEDED VTE, IMPEDE VTE, and SAVED Risk Models in Predicting Venous Thromboembolism in Multiple Myeloma Patients: A Retrospective Study in Türkiye"

_diagnostics, 2025, doi:10.3390/diagnostics15050633_

Round 1

Reviewer 1 Report

Comments and Suggestions for Authors

It is important to present data regarding VTE risk in MM even if they are retrospective and to evaluate the performance of current risk assessment models for VTE in MM in real-world populations (again even if the assessment is retrospective). I have a few major concerns however but I do believe these can be addressed by the authors to make the manuscript suitable for publication. 

1. I cannot see the tables/ figures which are necessary to allow an accurate reviewing process. 

2. The English language needs significant editing prior to publication. 

3. I would like to see data regarding mode of thromboprophylaxis among patients who had VTE events. In addition it is important to know what type of thromboprophylaxis patients received per risk category (as applied by the scores retrospectively), were patients receiving appropriate thromboprophylaxis based on their risk category? Did any patients receive prophylactic LMWH? 

4. What was the median time to event? Why did the authors chose to report the 6month cumulative incidence? What was median d-dimer level and what was the actual timing of the assessment? IT would also be important to know what was the VTE rate according to treatment type? 

5. How would the authors justify assessing the performance of the SAVED score which was specifically developed for patients on IMiDs in a non-IMID population?

6.Please see lines 225-227 “ We have also analyzed the performance of IMPEDED VTE score categorically for external validation, each increase in the IMPEDED VTE score was associated with (Table 4) “ Please complete the sentence 

Comments on the Quality of English Language

2. The English language needs significant editing prior to publication. 

Author Response

To begin with, on behalf of all authors, we would like to thank the reviewer for their effort and constructive comments. We believe that with their suggestions and comments, our manuscript will evolve.

Reviewer 1

Comment 1. I cannot see the tables/ figures which are necessary to allow an accurate reviewing process.

Answer 1 Frankly, we had five tables and four figures we did not understand why the reviewers could not see the tables. We will make the tables that the reviewers could see.  We have now attached all relevant tables and figures to ensure a thorough and accurate review process.

Comment 2. The English language needs significant editing prior to publication.

Answer 2 We have made necessary arrangements for English Grammar and punctuation to be able to read better. We have also refined the language and ensured the manuscript meets publication standards.

Comment 3. I would like to see data regarding mode of thromboprophylaxis among patients who had VTE events. In addition it is important to know what type of thromboprophylaxis patients received per risk category (as applied by the scores retrospectively), were patients receiving appropriate thromboprophylaxis based on their risk category? Did any patients receive prophylactic LMWH?

Answer 3. We appreciate the reviewer’s important question regarding thromboprophylaxis data. In our cohort, 215 patients (47.3%) were on low-dose acetylsalicylic acid (ASA), and 20 patients (4.4%) were on therapeutic low-molecular-weight heparin (LMWH) or vitamin K antagonists (VKAs). While we have this data, the small number of patients on therapeutic LMWH/VKAs (n=20) compared to those on ASA (n=215) makes it challenging to perform a meaningful comparison between the two groups. The disparity in group sizes limits our ability to draw robust conclusions about the relative effectiveness or appropriateness of thromboprophylaxis strategies in this retrospective analysis. Therefore we did not make a comparison between the two groups.

Comment 4. What was the median time to event? Why did the authors chose to report the 6month cumulative incidence? What was median d-dimer level and what was the actual timing of the assessment? IT would also be important to know what was the VTE rate according to treatment type?

Answer 4. The median time from multiple myeloma (MM) diagnosis to a venous thromboembolism (VTE) event was 97 days (range: 7–178 days). We have added this information to our manuscript. We chose the 6th-month cumulative incidence of thrombosis because both  IMPEDE VTE and IMPEDED scores were generated to show the 6th-month incidence of thrombosis. The median D-dimer value of the study participants was found as 1.21 µg/mL fibrinogen equivalent units (FEU), and the upper limit of the assay was 0.5s µg/mL FEU. Which was written on lines 206-207.Data regarding the association between VTE events and MM treatment regimens are summarized in Table 3. However, due to the heterogeneity of the treatment regimens used in our study, a direct comparison of VTE rates across different treatment types was not feasible.

Comment 5. How would the authors justify assessing the performance of the SAVED score which was specifically developed for patients on IMiDs in a non-IMID population?

Answer 5. The SAVED score was developed for IMiD-treated patients, we included it in our analysis to explore its generalizability to a broader MM population, including non-IMiD-treated patients. We have also wanted the clarify  the insights into its applicability across different treatment regimens.

Comment 6.Please see lines 225-227 “ We have also analyzed the performance of IMPEDED VTE score categorically for external validation, each increase in the IMPEDED VTE score was associated with (Table 4) “ Please complete the sentence

Answer 6. Thank you for bringing this up there was a typo error we have deleted the  last sentence and corrected it as follows “We have also analyzed the performance of the IMPEDED VTE score categorically for external validation. Each increase in the IMPEDED VTE score category was associated with a significantly higher risk of VTE: the 6-month cumulative incidence rates were 3.4% in the low-risk group (reference), 8.5% in the intermediate-risk group (HR 3.57, 95% CI 1.13–9.77), and 17.2% in the high-risk group (HR 12.89, 95% CI 11.33–19.75), as detailed in Table 4”

Reviewer 2 Report

Comments and Suggestions for Authors

The authors performed a comprehensive comparison and independent validaiton of two scores available for the evaluation of VTE risk in patients with multiple myeloma and show that the IMPEDED VTE score, which incorporates D-dimer as a biomarker, was found to outperform other risk models in predicting VTE in MM patients.

The external validation of VTE risk scores in patients with MM is required for the improvement of thromboprophylaxis in this field. The manuscript is well written and the data are comprehensively analysed

Author Response

To begin with, on behalf of all authors, we would like to thank the reviewer for their effort and constructive comments. We believe that with their suggestions and comments, our manuscript will evolve.

We thank the reviewer for their positive feedback and acknowledgment of the importance of our work in validating VTE risk scores in MM patients

Round 2

Reviewer 1 Report

Comments and Suggestions for Authors

I am happy with the authors responses and changes to the manuscript. Overall I would only recommend trying to slightly cut-down on the words as the manuscript is quite long. 

Author Response

 1-Overall I would only recommend trying to slightly cut-down on the words as the manuscript is quite long. 

thank you for your suggestions. Sentences and words have been reviewed. Possible abbreviations and arrangements have been made.